# SIAMESE CAPSULE NETWORKS

## ABSTRACT

Capsule Networks have shown encouraging results on *defacto* benchmark computer vision datasets such as MNIST, CIFAR and smallNORB. Although, they are yet to be tested on tasks where (1) the entities detected inherently have more complex internal representations and (2) there are very few instances per class to learn from and (3) where point-wise classification is not suitable. Hence, this paper carries out experiments on face verification in both controlled and uncontrolled settings that together address these points. In doing so we introduce *Siamese Capsule Networks*, a new variant that can be used for pairwise learning tasks. The model is trained using contrastive loss with $\ell_2$-normalized capsule encoded pose features. We find that *Siamese Capsule Networks* perform well against strong baselines on both pairwise learning datasets, yielding best results in the few-shot learning setting where image pairs in the test set contain unseen subjects.

## 1 INTRODUCTION

Convolutional Neural networks (CNNs) have been a mainstay model for a wide variety of tasks in computer vision. CNNs are effective at detecting local features in the receptive field, although the spatial relationship between features is lost when crude routing operations are performed to achieve translation invariance, as is the case with max and average pooling. Essentially, pooling results in viewpoint invariance so that small perturbations in the input do not effect the output. This leads to a significant loss of information about the internal properties of present entities (e.g location, orientation, shape and pose) in an image and relationships between them. The issue is usually combated by having large amounts of annotated data from a wide variety of viewpoints, albeit redundant and less efficient in many cases. As noted by hinton1985shape, from a psychology perspective of human shape perception, pooling does not account for the coordinate frames imposed on objects when performing mental rotation to identify handedness Rock (1973); McGee (1979); Humphreys (1983). Hence, the scalar output activities from local kernel regions that summarize sets of local inputs are not sufficient for preserving reference frames that are used in human perception, since viewpoint information is discarded. Spatial Transformer Networks (STN) Jaderberg et al. (2015) have acknowledged the issue by using dynamic spatial transformations on feature mappings to enhance the geometric invariance of the model, although this approach addresses changes in viewpoint by learning to remove rotational and scale variance, as opposed to viewpoint variance being reflected in the model activations. Instead of addressing translation invariance using pooling operations, Hinton et al. (2011) have worked on achieving translation equivariance.The recently proposed Capsule Networks Sabour et al. (2017); Hinton et al. (2018) have shown encouraging results to address these challenges. Thus far, Capsule Networks have only been tested on datasets that have (1) a relatively sufficient number of instances per class to learn from and (2) utilized on tasks in the standard classification setup. This paper extends Capsule Networks to the pairwise learning setting to learn relationships between whole entity encodings, while also demonstrating their ability to learn from little data that can perform few-shot learning where instances from new classes arise during testing (i.e zero-shot prediction). The *Siamese Capsule Network* is trained using a contrastive loss with $\ell_2$-normalized encoded features and demonstrated on two face verification tasks.

## 2 CAPSULE NETWORKS

Hinton et al. (2011) first introduced the idea of using whole vectors to represent internal properties (referred to as instantiation parameters that include pose) of an entity with an associated activation probability where each capsule represents a single instance of an entity within in an image. This differs from the single scalar outputs in conventional neural networks where pooling is used as a crude routing operation over filters. Pooling performs sub-sampling so that neurons are invariant to viewpoint change, instead capsules look to preserve the information to achieve equivariance, akin to perceptual systems. Hence, pooling is replaced with a dynamic routing scheme to send lower-level capsule (e.g nose, mouth, ears etc.) outputs as input to parent capsule (e.g face) that represent part-whole relationships to achieve translation equivariance and untangles the coordinate frame of an entity through linear transformations. The idea has its roots in computer graphics where images are rendered given an internal hierarchical representation, for this reason the brain is hypothesized to solve an *inverse graphics* problem where given an image the cortex deconstructs it to its latent hierarchical properties. The original paper by Sabour et al. (2017) describes a dynamic routing scheme that represent these internal representations as vectors given a group of designated neurons called *capsules*, which consist of a pose vector $u \in \mathbb{R}^d$ and activation $\alpha \in [0, 1]$. The architecture consists of two convolutional layers that are used as the initial input representations for the first capsule layer that are then routed to a final class capsule layer. The initial convolutional layers allow learned knowledge from local feature representations to be reused and replicated in other parts of the receptive field. The capsule inputs are determined using a *Iterative Dynamic Routing* scheme. A transformation $W_{ij}$ is made to output vector $u_i$ of capsule $C_i^L$. The length of the vector $u_i$ represents the probability that this lower-level capsule detected a given object and the direction corresponds to the state of the object (e.g orientation, position or relationship to upper capsule). The output vector $u_i$ is transformed into a prediction vector $\hat{u}_{j|i}$, where $\hat{u}_{j|i} = W_{ij}u_i$. Then, $\hat{u}_{j|i}$ is weighted by a coupling coefficient $c_{ij}$ to obtain $s_j = \sum_i c_{ij}\hat{u}_{j|i}$, where coupling coefficients for each capsule $\sum_j c_{ij} = 1$ and $c_{ij}$ is got by log prior probabilities $b_{ij}$ from a sigmoid function, followed by the `softmax`, $c_{ij} = e^{b_{ij}} / \sum_k e^{b_{ik}}$. If $\hat{u}_{j|i}^L$ has high scalar magnitude when multiplied by $u_j^{L+1}$ then the coupling coefficient $c_{ij}$ is increased and the remaining potential parent capsules coupling coefficients are decreased. *Routing By Agreement* is then performed using coincidence filtering to find tight clusters of nearby predictions. The entities output vector length is represented as the probability of an entity being present by using the nonlinear normalization shown in Equation 1 where vote $v_j$ is the output from total input $s_j$, which is then used to compute the agreement $a_{ij} = v_j\hat{u}_{j|i}$ that is added to the log prior $b_{ij}$.

$$\mathbf{v}_j = \frac{||\mathbf{s}_j||^2}{1 + ||\mathbf{s}_j||^2} \frac{\mathbf{s}_j}{||\mathbf{s}_j||} \tag{1}$$

The capsule is assigned a high log-likelihood if densely connected clusters of predictions are found from a subset of $s$. The centroid of the dense cluster is output as the entities generalized pose. This coincidence filtering step can also be achieved by traditional outlier detection methods such as Random sample consensus (RANSAC) Fischler & Bolles (1987) and classical Hough Transforms Ballard (1987) for finding subsets of the feature space with high agreement. Although, the motivation for using the vector normalization of the instantiation parameters is to force the network to preserve orientation. Lastly, a reconstruction loss on the images was used for regularization which constrains th capsules to learn properties that can better encode the entities. In this paper, we do not use such regularization scheme by autoencoding pairs of input images, instead we use a variant of dropout.

**Extensions Of Capsule Networks**     Hinton et al. (2018) recently describe matrix capsules that perform routing by agreement using the expectation maximization (EM) algorithm, motivated by computer graphics where pose matrices are used to define rotations and translations of objects to account for viewpoint changes. Each parent capsule is considered a Gaussian and the pose matrix of each child capsule are considered data samples of the Gaussian. A given layer $L$ contains a set of capsules $C^L$ such that $\forall C^\ell \exists \{\mathcal{M}^\ell, \alpha^\ell\} \in C^L$ where pose matrix $\mathcal{M}^\ell \in \mathbb{R}^{n \times n}$ ($n = 4$) and activation $\alpha^\ell \in [0, 1]$ are the outputs. A vote is made $V_{ij}^\ell = \mathcal{M}_i^\ell W_{ij}^\ell$ for the pose matrix of $C_j^{L+1}$ where $W_{ij} \in \mathbb{R}^{n \times n}$ is a learned viewpoint invariant transformation matrix from capsule

$C_i^L \rightarrow C_j^{L+1}$. EM determines the activation of $C_j^{L+1}$ as $a_j = \sigma\big(\lambda(\beta_a - \beta_u \sum_i r_{ij} - \sum_h cost_j^h)\big)$ where the $cost_j^h$ is the negative log-probability density weighted by the assignment probabilities $r_{ij}$, $-\beta_u$ is the negative log probability density per pose matrix computed to describe $C_j^{L+1}$. If $C_j^{L+1}$ is activated $-\beta_a$ is the cost for describing $(\mu_j, \sigma_j^2)$ from lower-level pose data samples along with $r_{ij}$ and $\lambda$ is the inverse temperature so as the assignment probability becomes higher the slope of the sigmoid curve becomes steeper (represents the presence of an entity instead of the nonlinear vector normalization seen in Equation 1). The network uses 1 standard convolutional layer, a primary capsule layer, 2 intermediate capsule convolutional layer, followed by the final class capsule layer. The matrix capsule network significantly outperformed CNNs on the SmallNORB dataset.

LaLonde & Bagci (2018) introduce SegCaps which uses a locally connected dynamic routing scheme to reduce the number of parameters while using deconvolutional capsules to compensate for the loss of global information, showing best performance for segmenting pathological lungs from low dose CT scans. The model obtained a 39% and 95% reduction in parameters over baseline architectures while outperforming both.

Bahadori (2018) introduced Spectral Capsule Networks demonstrated on medical diagnosis. The method shows faster convergence over the EM algorithm used with pose vectors. Spatial coincidence filters align extracted features on a 1-d linear subspace. The architecture consists of a 1d convolution followed by 3 residual layers with dilation. Residual blocks $R$ are used as nonlinear transformations for the pose and activation of the first primary capsule instead of the linear transformation that accounts for rotations in CV, since deformations made in healthcare imaging are not fully understood. The weighted votes are obtained as $s_{j,i} = \alpha_i R_j(u_i) \quad \forall i$ where $S_j$ is a matrix of concatenated votes that are then decomposed using SVD, where the first singular value dimension $\tilde{s}_1$ is used to capture most of the variance between votes, thus the activation $a_j$ activation is computed as $\sigma\big(\eta(s_1^2 / \sum_k s_k^2 - b)\big)$ where $s_1^2 / \sum_k s_k^2$ is the ratio of all variance explained for all right singular vectors in $V$, $b$ is optimized and $\eta$ is decreased during training. The model is trained by maximizing the log-likelihood showing better performance than the spread loss used with matrix capsules and mitigates the problem of capsules becoming dormant.

Wang & Liu (2018) formalize the capsule routing strategy as an optimization of a clustering loss and a KL regularization term between the coupling coefficient distribution and its past states. The proposed objective function follows as $\min_{C,S}\{\mathcal{L}(C,S) := -\sum_i \sum_j c_{ij}\langle o_{j|i}, s_j\rangle + \alpha \sum_i \sum_j c_{ij} \log c_{ij}\}$ where $o_{j|i} = T_{ij}\mu_i / ||T_{ij}||_\mathcal{F}$ and $||T_{ij}||_\mathcal{F}$ is the Frobenious norm of $T_{ij}$. This routing scheme shows significant benefit over the original routing scheme by Sabour et al. (2017) as the number of routing iterations increase. Evidently, there has been a surge of interest within the research community.

In contrast, the novelty presented in this paper is the pairwise learning capsule network scheme that proposes a different loss function, a change in architecture that compares images, aligns entities across images and describes a method for measuring similarity between final layer capsules such that inter-class variations are maximized and intra-class variations are minimized. Before describing these points in detail, we briefly describe the current state of the art work (SoTA) in face verification that have utilized Siamese Networks.

## 3 SIAMESE NETWORKS FOR FACE VERIFICATION

Siamese Networks (SNs) are neural networks that learn relationships between encoded representations of instance pairs that lie on low dimensional manifold, where a chosen distance function $d_\omega$ is used to find the similarity in output space. Below we briefly describe state of the art convolutional SN's that have been used for face verification and face recognition.

Sun et al. (2014) presented a joint identification-verification approach for learning face verification with a contrastive loss and face recognition using cross-entropy loss. To balance loss signals for both identification and verification, they investigate the effects of varying weights controlled by $\lambda$ on the intra-personal and inter-personal variations, where $\lambda = 0$ leaves only the face recognition loss and $\lambda \rightarrow \infty$ leaves the face verification loss. Optimal results are found when $\lambda = 0.05$ intra personal variation is maximized while both class are distinguished.

Wen et al. (2016) propose a center loss function to improve discriminative feature learning in face recognition. The center loss function proposed aims to improve the discriminability between feature representations by minimizing the intra-class variation while keeping features from different classes separable. The center loss is given as $\mathcal{L} = -\sum_{i=1}^{m} \log(e^z)/(\sum_{j=1}^{n} e^z) + \lambda_2 \sum_m i = 1||x_i - c_{y_i}||_2^2$ where $z = W_j^T x_i + b_j$. The $c_{y_i}$ is the centroid of feature representations pertaining to the $i^{th}$ class. This penalizes the distance between class centers and minimizes the intra-class variation while the `softmax` keeps the inter-class features separable. The centroids are computed during stochastic gradient descent as full batch updates would not be feasible for large networks.

Liu et al. (2017) proposed *Sphereface*, a hypersphere embedding that uses an angular softmax loss that constrains disrimination on a hypersphere manifold, motivated by the prior that faces lie on a manifold. The model achieves 99.22 % on the LFW dataset, and competitive results on Youtube Face (YTF) and MegaFace. Sankaranarayanan et al. (2016) proposed a triplet similarity embedding for face verification using a triple loss $\arg\min_W = \sum_{\alpha,p,n \in T} \max(0, \alpha + \alpha^T W^T W(n-p))$ where for $T$ triplet sets lies an anchor class $\alpha$, positive class $p$ and negative class $n$, a projection matrix $W$, (performed PCA to obtain $W_0$) is minimized with the constraint that $W_a^T W_p > W_a^T W_n$. The update rule is given as $W_{t+1} = W_t - \eta W_t(\alpha(n-p)^T + (n-p)\alpha^T)$. Hu et al. (2014) use deep metric learning for face verification with loss $\arg\min_f J = \frac{1}{2}\sum_{i,j} g(1 - \ell_{i,j}(\tau - d_f^2(x_i, x_j))) + \frac{\lambda}{2}(\sum_{m=1}^{M}(|\theta^{(m)}|_F^2))$ where $g(z) = \log(1 + e^{\beta z})/\beta$, $\beta$ controls the slope steepness of the logistic function, $||A||_{\mathcal{F}}$ is the frobenius norm of $A$ and $\lambda$ is a regularization parameter. Hence, the loss function is made up of a logistic loss and regularization on parameters $\theta = [W, b]$. Best results are obtained using a combination of SIFT descriptors, dense SIFT and local binary patterns (LBP), obtaining 90.68% (+/- 1.41) accuracy on the LFW dataset.

Ranjan et al. (2017) used an $\ell_2$-constraint on the softmax loss for face verification so that the encoded face features lie on the ambit of a hypersphere, showing good improvements in performance. This work too uses an $\ell_2$-constraint on capsule encoded face embeddings.FaceNet Schroff et al. (2015) too uses a triplet network that combines the Inception network Szegedy et al. (2015) and a 8-layer convolutional model Zeiler & Fergus (2014) which learns to align face patches during training to perform face verification, recognition and clustering. The method trains the network on triplets of increasing difficulty using a negative example mining technique. Similarly, we consider a Siamese Inception Network for the tasks as one of a few comparisons to SCNs.

The most relevant and notable use of Siamese Networks for face verification is the *DeepFace* network, introduced by Taigman et al. (2014). The performance obtained was on par with human level performance on the *Faces in the Wild* (LFW) dataset and significantly outperformed previous methods. However, it is worth noting this model is trained on a large dataset from Facebook (SFC), therefore the model can be considered to be performing transfer learning before evaluation. The model also carries out some manual steps for detecting, aligning and cropping faces from the images. For detecting and aligning the face a 3D model is used. The images are normalized to avoid any differences in illumination values, before creating a 3D model which is created by first identifying 6 fiducial points in the image using a Support Vector Regressor from a LBP histogram image descriptor. Once the faces are cropped based on these points, a further 67 fiducial point are identified for 3D mesh model, followed by a piecewise affine transformation for each section of the image. The cropped image is then passed to 3 CNN layers with an initial max-pooling layer followed two fully-connected layers. Similar to Capsule Networks, the authors refrain from using max pooling at each layer due to information loss. In contrast to this work, the only preprocessing steps for the proposed SCNs consist of pixel normalization and a reszing of the image.

The above work all achieve comparable state of the art results for face verification using either a single CNN or a combination of various CNNs, some of which are pretrained on large related datasets. In contrast, this work looks to use a smaller Capsule Network that is more efficient, requires little preprocessing steps (i.e only a resizing of the image and normalization of input features, no aligning, cropping etc.) and can learn from relatively less data.

## 4 SIAMESE CAPSULE NETWORK

The Capsule Network for face verification is intended to identify enocded part-whole relationships of facial features and their pose that in turn leads to an improved similarity measure by aligning

capsule features across paired images. The architecture consists of a 5-hidden layer (includes 2 capsule layers) network with tied weights (since both inputs are from the same domain). The $1^{st}$ layer is a convolutional filter with a stride of 3 and 256 channels with kernels $\kappa_i^1 \in \mathbb{R}^{9\times9} \ \forall i$ over the image pairs $\langle x_1, x_2 \rangle \in \mathbb{R}^{100\times100}$ , resulting in 20, 992 parameters. The $2^{nd}$ layer is the primary capsule layer that takes $\kappa^{(1)}$ and outputs $\kappa^{(2)} \in \mathbb{R}^{31\times31}$ matrix for 32 capsules, leading to $5.309 \times 10^6$ parameters (663, 552 weights and 32 biases for each of 8 capsules). The $3^{rd}$ layer is the face capsule layer, representing the routing of various properties of facial features, consisting of $5.90 \times 10^6$ parameters. This layer is then passed to a single fully connected layer by concatenating the pose vectors $\mathcal{M}_\cap^L = \cap_{i=1}^{|C^L|}$ as input, while the sigmoid functions control the dropout rate for each capsule during training. The nonlinear vector normalization shown in Equation 1 is replaced with a tanh function $\tanh(.)$ which we found in initial testing to produce better results. Euclidean distance, Manhattan distance and cosine similarity are considered as measures between the capsule image encodings. The aforementioned SCN architecture describes the setup for the AT&T dataset. For the LFW dataset, 6 routing iterations are used and 4 for AT&T.

**Capsule Encoded Representations**    To encode paired images $\langle x_1, x_2 \rangle$ into vector pairs $\langle h_1, h_2 \rangle$ the pose vector of each capsule is vectorized and passed as input to a fully connected layer containing 20 activation units. Hence, for each input there is a lower 20-dimensional representation of 32 capsule pose vectors resulting in 512 input features. To ensure all capsules stay active the dropout probability rate is learned for each capsule. The sigmoid function learns the dropout rate of the final capsule layer using *Concrete Dropout* Gal et al. (2017), which builds on prior work Kingma et al. (2015); Molchanov et al. (2017) by using a continuous relaxation that approximates the discrete Bernoulli distribution used for dropout, referred to as a concrete distribution. Equation 2 shows the objective function for updating the concrete distribution. For a given capsule probability $p_c$ in the last capsule layer, the `sigmoid` computes the relaxation $\tilde{z}$ on the Bernoulli variable $z$, where $u$ is drawn uniformly between [0,1] where $t$ denotes the temperature values ($t = 0.1$ in our experiments) which forces probabilities at the extremum when small. The pathwise derivative estimator is used to find a continuous estimation of the dropout mask.

$$\tilde{z}_t = \sigma\Big(\frac{1}{t}\big(\log p_c - \log(1 - p_c)\big) + \log u_c - \log(1 - u_c)\Big) \tag{2}$$

**Loss Functions**    The original capsule paper with dynamic routing Sabour et al. (2017) used a margin loss $L_c = T_c \max(0, m^+ - ||v_c||)^2 + \lambda(1 - T_c)\max(0, ||v_c|| - m^-)^2$ where the class capsule $v_c$ has margin $m^+ = 0.9$ positives and $m^- = 1 - m^+$ negatives. The weight $\lambda$ is used to prevent the activity vector lengths from deteriorating early in training if a class capsule is absent. The overall loss is then simply the sum of the capsule losses $\sum_c L_c$. A spread loss Hinton et al. (2018) has also been used to maximize the inter-class distance between the target class and the remaining classes for classifying on the smallNORB dataset. This is given as $L_i = (\max(0, m - (a_t - a_i))^2, \quad L = \sum_{i\neq t} L_i$ where the margin $m$ is increased linearly during training to ensure lower-level capsule stay active throughout training. This work instead uses a contrastive margin loss Chopra et al. (2005) where the aforementioned capsule encoding similarity function $d_\omega$ outputs a predicted similarity score. The contrastive loss $\mathcal{L}_c$ ensures similar vectorized pose encodings are drawn together and dissimilar poses repulse. Equation 3 shows a a pair of images that are passed to the SCN model where $D_w = ||f^\omega(x_1) - f^\omega(x_2)||_2^2$ computes the Euclidean distance between encodings and $m$ is the margin. When using Manhattan distance $D_w = \exp\big(-||f^\omega(x_1) - f^\omega(x_2)||_1\big)$ in which case $m \in [0, 1)$. is used where $y \in [-1, 1]$.

$$L_c(\omega) = \sum_{i=1}^{m} \Big(\frac{1}{2}(1 - y^{(i)})D_\omega^{(i)} + \frac{1}{2}y^{(i)}max(0, m - D_\omega^{(i)})\Big) \tag{3}$$

A double margin loss that has been used in prior work by Lin et al. (2015) is also considered to affect matching pairs such that to account for positive pairs that can also have high variance in the distance measure. It is worth noting this double margin is similar to the aforementioned margin loss used on class capsules, without the use of $\lambda$. Equation 4 shows the double-margin contrastive loss where positive margin $m_p$ and negative margin $m_n$ are used to find better separation between matching

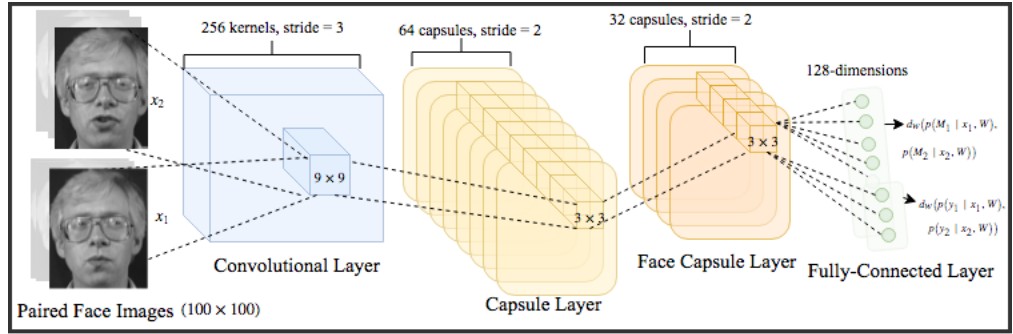

Figure 1: *Siamese Capsule Network Architecture*

and non-matching pairs. This loss is only used for LFW, given the limited number of instances in AT&T we find the amount of overlap between pairs to be less severe in experimentation.

$$L_c(\omega) = \sum_{i=1}^{m} \left( (1 - y^{(i)}) max(0, D_\omega^{(i)} - m_n)^2 + y^{(i)} max(m_p - D_\omega^{(i)}, 0)^2 \right) \quad (4)$$

The original reconstruction loss $L_r(\theta) = \sum_{j=1}^{2} \sum_{i=1}^{m} (y_i^{(j)} - \hat{y}_i^{(j)})^2$ used as regularization is not used in the pairwise learning setting, instead we rely on the dropout for regularization with exception of the SCN model that uses concrete dropout on the final layer.

**Optimization**   Convergence can often be relatively slow for face verification tasks, where few informative batch updates (e.g a sample with significantly different pose for a given class) get large updates but soon after the effect is diminished through gradient exponential averaging (originally introduced to prevent $\alpha \to 0$). Motivated by recent findings that improve adaptive learning rates we use AMSGrad  Reddi et al. (2018). AMSGrad improves over ADAM in some cases by replacing the exponential average of squared gradients with a maximum that mitigates the issue by keeping long-term memory of past gradients. Thus, AMSGrad does not increase or decrease the learning rate based on gradient changes, avoiding divergent or vanishing step sizes over time. Equation 5 presents the update rule, where diagonal of gradient $g_t$ is given as $v_t = \Theta_2 v_{t-1} + (1 - \Theta_2) g_t^2$, $m_t = \Theta_1 m_{t-1} + (1 - \Theta_1) g_t$, $\alpha_t = 1/\sqrt{t}$, $\hat{v}_t = max(\hat{v}_{t-1}, v_t)$, ensuring $\alpha$ is monotonic.

$$\omega_{t+1} = \omega_t - \alpha \frac{m_t}{\sqrt{\hat{v}_t + \epsilon}} \quad (5)$$

## 5   EXPERIMENTS ON FACE VERIFICATION

**A. AT&T dataset**   The AT&T face recognition and verification dataset consists of 40 different subjects with only 10 gray-pixel images per subject in a controlled setting. This smaller dataset allows us to test how SCNs perform with little data. For testing, we hold out 5 subjects so that we are testing on unseen subjects, as opposed to training on a given viewpoint of a subject and testing on another viewpoint of the same subject. Hence, zero-shot pairwise prediction is performed during testing.

**B. Labeled Faces In The Wild (LFW) dataset**   The LFW consists of 13,000 colored photographed faces from the web. This dataset is significantly more complex not only because there 1680 subjects, with some subjects only consisting of two images, but also because of varied amount of aging, pose, gender, lighting and other such natural characteristics. Each image is $250 \times 250$, in this work the image is resized to $100 \times 100$ and normalized. From the original LFW dataset there has been 2 different versions of the dataset that align the images using funneling  Huang et al. (2007) and deep funneling  Huang et al. (2012). The latter learns to align the images using Restricted Boltzmann Machines with a group sparsity penalty, showing performance improvements for face

| | AT&T | | LFW | | LFW+Double-M | |
|---|---|---|---|---|---|---|
| Models | Train | Test | Train | Test | Train | Test |
| Standard | 0.013 | 0.042 | 0.0021 | 0.012 | 0.0049 | 0.014 |
| ResNet-34 | 0.015 | 0.057 | 0.0018 | 0.012 | 0.0026 | 0.013 |
| AlexNet | 0.032 | 0.085 | 0.0019 | 0.009 | 0.0021 | 0.010 |
| SCNet | 0.008 | 0.019 | 0.0020 | 0.013 | 0.0019 | 0.011 |
| SDropCapNet | 0.010 | 0.032 | 0.0023 | 0.010 | 0.0028 | 0.012 |

Table 1: 5-fold CV Train & Test Contrastive Loss w/ Malahaobonis distance

verification tasks. The penalty leads to an arrangement of the filters that improved the alignment results. This overcomes the problems previous CNNs and models alike had in accounting for pose, orientation and problems Capsule Networks look to address. In contrast, we use the original raw image dataset.

Both allow for a suitable variety as the former only contains grey-pixel images, a smaller dataset with very few instances per class and images taken in a constrained setting allowing for a more refined analysis, while the LFW data samples are colored images, relatively large with unbalanced classes and taken in an unconstrained setting.

**Baselines** SCNs are compared against well-established architectures for image recognition and verification tasks, namely AlexNet, ResNet-34 and InceptionV3 with 6 inception layers instead of the original network that uses 8 layers which are used many of the aforementioned papers in Section 3.

## 5.1 RESULTS

Table 1 shows best test results obtained when using contrastive loss with Euclidean distance between encodings (i.e Mahalanobis distance) for both $AT\&T$ and LFW over 100 epochs. The former uses $m = 2.0$ and the latter uses $m = 0.2$, while for the double margin contrastive loss $m_n = 0.2$ matching margin and $m_p = 0.5$ negative matching margin is selected. These settings were chosen during 5-fold cross validation, grid searching over possible margin settings. SCN outperforms baselines on the $AT\&T$ dataset after training for 100 epochs. We find that because AT&T contains far fewer instances an adapted dropout rate leads to a slight increase in contrastive loss. Additionally, adding a reconstruction loss with $\lambda_r = 1e^{-4}$ for both paired images led to a decrease in performance when compared to using dropout with a rate $p = 0.2$ on all layers except the final layer that encodes the pose vectors. We find for the LFW dataset that the SCN and AlexNet have obtained the best results while SCN has 25% less parameters. Additionally, the use of a double margin results in better results for the standard SCN but a slight drop in performance when used with concrete dropout on the final layer (i.e SDropCapNet).

Figure 2 illustrates the contrastive loss during training $\ell_2$-normalized features for each model tested with various distance measures on AT&T and LFW. We find that SCN yields faster convergence on AT&T, particularly when using Manhattan distance. However for Euclidean distance, we observe a loss variance reduction during training and the best overall performance. Through experiments we find that batch normalized convolutional layers improves performance of the SCN. In batch normalization, $\hat{x}^{(k)} = (x^{(k)} - \mathbb{E}[x^k])/\sqrt{Var[x^{(k)}]}$ provides a unit Gaussian batch that is shifted by $\gamma^{(k)}$ and scaled with $\beta^{(k)}$ so that $a^{(k)} = \gamma^{(k)}\hat{x}^{(k)} + \beta^{(k)}$. This allows the network to learn whether the input range should be more or less diffuse. Batch normalization on the initial convolutional layers reduced variance in loss during training on both the $AT\&T$ and $LFW$ datasets. LFW test results show that the SCN model takes longer to converge particularly in the early stages of training, in comparison to AlexNet.

Figure 3 shows the probability density of the positive pair predictions for each model for all distances between encodings with contrastive loss for the LFW dataset. We find the variance of predictions is lower in comparison to the remaining models, showing a higher precision in the predictions, particularly for Manhattan distance. Additionally, varying distances for these matching images were close

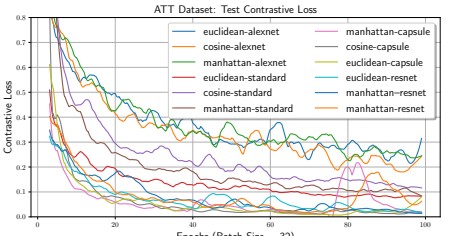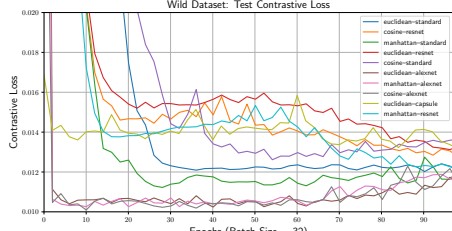

Figure 2: Contastive Loss for AT&T (left) and LFW (right) datasets

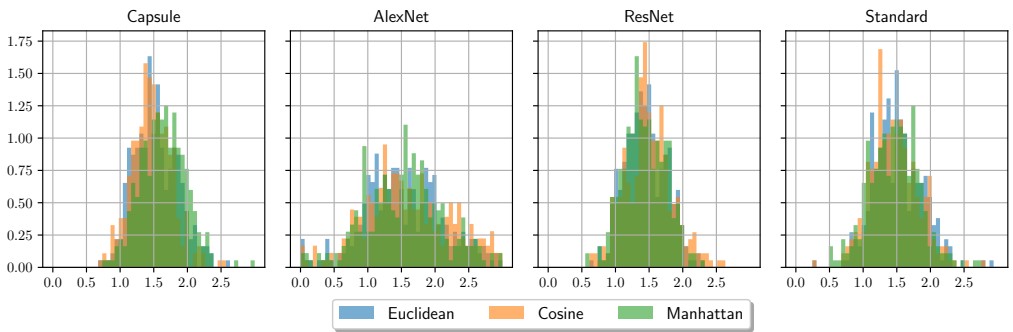

Figure 3: *Probability Density of LFW Positive Pair Test Predictions*

in variance to non-matching images. This motivated the use of the double margin loss considered for the LFW dataset.

Finally, the SCN model has between 104-116 % less parameters than Alexnet, 24-27 % Resnet-34 and 127-135% less than the best standard baseline for both datasets. However, even considering tied weights between models in the SCN, Capsule Networks are primarily limited in speed even with a reduction in parameters due to the routing iterations that are necessary during training.

## 6  CONCLUSION

This paper has introduced the *Siamese Capsule Network*, a novel architecture that extends Capsule Networks to the pairwise learning setting with a feature $\ell_2$-normalized contrastive loss that maximizes inter-class variance and minimizes intra-class variance. The results indicate Capsule Networks perform better at learning from only few examples and converge faster when a contrastive loss is used that takes face embeddings in the form of encoded capsule pose vectors. We find *Siamese Capsule Networks* to perform particularly well on the AT&T dataset in the few-shot learning setting, which is tested on unseen classes (i.e subjects) during testing, while competitive against baselines for the larger *Labeled Faces In The Wild* dataset.

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
