# OpenReview forum: "Siamese Capsule Networks"
_ICLR.cc/2019/Conference_

### Official Review · AnonReviewer2 · 2018-10-30
**Good direction for research on capsules, but results too weak and idea too incremental**

**Rating:** 3
**Confidence:** 4

**Review:**

This paper presents an extension of Capsule Networks, Siamese Capsule Networks (SCNs), that can be applied to the problem of face verification. Results are reported on the small AT&T dataset and the LFW dataset.

I like the direction that this paper is taking. The original Capsules work has been looking at fairly simple and small scale datasets, and the natural next step for this approach is to start addressing harder datasets, LFW being one of them. Also face verification is a natural problem to look at with Capsules.

However, I think this paper currently falls short of what I would expect from an ICLR paper. First, the results are not particularly impressive. Indeed, SCN doesn't outperform AlexNet on LFW (the most interesting dataset in the experiments). Also, I'm personally not particularly compelled by the use of the contrastive loss as the measure of performance, as it is sensitive to the scaling of the particular representation f(x) used to compute distances. Looking at accuracy (as in other face verification papers, such as DeepFace) for instance would have been more appropriate, in my opinion. I'm also worried about how hyper-parameters were selected. There are A LOT of hyper-parameters involved (loss function hyper-parameters, architecture hyper-parameters, optimizer hyper-parameters) and not much is said about how these were chosen. It is mentioned that cross validation was used to select some margin hyper-parameters, but results in Table 1 are also cross-validation results, which makes me wonder whether hyper-parameters were tuned on the performance reported in Table 1 (which of course would be biased).

The paper is also pretty hard to read. I recognize that there is a lot of complicated literature to cover (e.g. prior work on Capsule Networks has introduced variations on various aspects which are each complicated to describe). But as it currently reads, I can honestly say that I'm not 100% sure what exactly was implemented, i.e. which components of previous Capsule Networks were actually used in the experiments and which weren't. For example, I wasn't able to figure out which routing mechanism was used in this paper. The paper would strongly benefit from more explicitly laying out the exact definition of SCN, perhaps at the expense of enumerating all the other variants of capsules and losses that previous work has used.

Finally, regardless of the clarify of the paper, the novelty in extending Capsule Networks to a siamese architecture is arguably pretty incremental. This wouldn't be too much of a problem if the experimental results were strong, but unfortunately it isn't the case.

In summary:

Pros
- New extension of Capsule Networks, tackling a more challenging problem than previous work

Cons
- Novelty is incremental
- Paper lacks clarity and is hard to read
- Results are underwhelming

For these reasons, I'm afraid I can't recommend this paper be accepted.

Finally, I've noted the following typos:
- hinton1985shape => use proper reference
- within in => within
- that represent => that represents
- a Iterated => an Iterated
- is got => is obtained
- followed two => followed by two
- enocded => encoded
- a a pair => a pair
- such that to => such as to
- there 1680 subjects => there are 1680 subjects
- of varied amount => of the varied amount
- are used many => are used in many
- across the paper: lots of in-text references should be in parenthesis

---

### Official Review · AnonReviewer1 · 2018-11-04
**This paper has some new ideas about Capsule Network**

**Rating:** 6
**Confidence:** 4

**Review:**

In this paper, the author extends Capsule Network on the task of face verification to solve the problem of learning from only few examples and speeding the convergence. They propose a Siamese Capsule Network, which extends Capsule Networks to the pairwise learning setting with a feature l2-normalized contrastive loss that maximizes inter-class variance and minimizes intra-class variance. Here is a list of suggestions that will help the authors to improve this paper.
1.	The pairwise learning setting allow learning relationships between whole entity encodings
2.	The ability to learn from little data that can perform few-shot learning where instances from new classes arise during testing
3.	When a contrastive loss is used that takes face embeddings in the form of encoded capsule pose vectors, speed of converging is lifted.
4.	The description of experiment is too brief to show specific details.
5.	The figure of Siamese Capsule Network Architecture (figure 1) cannot show kernel of author(s)’s method, and lack explanation in the paper.

---

> ### Author Response · Authors · 2018-11-13
> **Author Response**
>
> Thanks for reviewing the paper.
>
> 5 - I will try to fit more training details
> 6 - A revised diagram will be put into the update version.

---

### Official Review · AnonReviewer3 · 2018-11-04
**Limited Contribution, Unclear results**

**Rating:** 5
**Confidence:** 4

**Review:**

Authors present an adaptation of Capsule Networks for pairwise learning tasks. The pose vectors of final capsule layers for each tower is concatenated and passed through a fully connected layer to calculate the embedding for each tower's input. Then a contrastive loss based on a distance metric is optimized for the embeddings. An interesting regularizer (?) is used which is a dropout based on Capsule activation for connecting last layer capsules to the fully connected layer.

Pros:

The literature review is rich and complete. In general authors explain details of the previous techniques as they use them too which is a good writing technique and improves the readability.

By utilizing Capsules authors avoid a rigorous preprocessing as it is common with the community. As I understand they do not even use face landmarks to align images.

Measured by the optimized loss, the proposed method achieves significant improvement upon baseline in the small At&t dataset.

Cons:

The contribution of this work is not on par with ICLR standard for conference papers. Specially since SDropCapsNet (the added dropout) is seems to be auxiliary (gives a slight boost only in LFW without double margin).

The method used for reporting results is unjustified and not compareable to prior work. For face verification, identification one should report at least an ROC curve based on a threshold on the distance or nearest neighbor identification results which are standards in the literature. Where as they only report the contrastive loss of their model and their own implementation of baselines and Figure 3 which does not clearly show any advantage for CapsNet Siamese networks.


Question:
The architecture description for last layer is vague. In text 512 is mentioned as the input dimmension, 512 is 16*32, Figure 1 shows 9 features per capsule or 3x3 kernels over capsules where it has to be fully connected? Also it says last layer is 128 dimmension where the text implies it should be 20. Could you please explain the role of 20?

Is table 1 the final contrastive loss achievable for each model?

Have you tried just gating the pose parameters of last layer by their activation (multiply to the tanh) rather than using a stochastic dropout?

---

> ### Author Response · Authors · 2018-11-13
> **Author Response**
>
> Yes, you are right we do not use face landmarks to align images. We instead rely on the final fully connected layer to carry this out from the flattened pose vectors.
> I also have taken on board your comment about using Receive Operator Curve for evaluation, which will be in an updated version soon.
>
> Q1. Yes apologies for the confusion. So there are 20 capsules with a dimension from 16 output channels in the last layer. The weights for the last fully connected layer are then W \in \mathbb{R}^{320x128}. I forgot to update the description and I will also amend the diagram to show that there is 20 capsules which output 16 dimensions each.
>
>
> Q2.  Table 1 is the contrastive loss for each model. Equation 3 used for AT&T and double margin loss from Equation 4 for LFW results.
>
> Q3. I did try something like this. I used a weighted average between each pose vector where the weights are set by normalising pose activation parameters in 0-1, but it did not perform as well as using a fully-connected layer with dropout (with or without a learnable dropout rate).

---

### Meta-Review · Area_Chair1 · 2018-12-12
**Incremental extension of capsule networks with results on face verification**

**Confidence:** 4
**Recommendation:** Reject

**Metareview:**

The paper extends capsule networks with a pairwise learning objective and evaluates on small face verification datasets. The authors do a great job describing prior work, but lack clarity when articulating their contribution and proposed method. In addition, some important implementation details, such as hyperparameter selection, are missing causing further confusion as to the final approach. Overall, according to the experiments shown, the approach offers modest improvements over prior work.

The approach offers an interesting and promising direction. We encourage the authors to revise the manuscript to clarify their approach and contribution and to improve their evaluation by including the relevant metrics and implementation details.